# The Combination of a Human Biomimetic Liver Microphysiology System with BIOLOGXsym, a Quantitative Systems Toxicology (QST) Modeling Platform for Macromolecules, Provides Mechanistic Understanding of Tocilizumab- and GGF2-Induced Liver Injury

**DOI:** 10.3390/ijms24119692

**Published:** 2023-06-02

**Authors:** James J. Beaudoin, Lara Clemens, Mark T. Miedel, Albert Gough, Fatima Zaidi, Priya Ramamoorthy, Kari E. Wong, Rangaprasad Sarangarajan, Christina Battista, Lisl K. M. Shoda, Scott Q. Siler, D. Lansing Taylor, Brett A. Howell, Lawrence A. Vernetti, Kyunghee Yang

**Affiliations:** 1DILIsym Services Division, Simulations Plus Inc., Research Triangle Park, Durham, NC 27709, USAscott.siler@simulations-plus.com (S.Q.S.);; 2Department of Computational and Systems Biology, Drug Discovery Institute, University of Pittsburgh, Pittsburgh, PA 15219, USAgough@pitt.edu (A.G.); dltaylor@pitt.edu (D.L.T.); 3Metabolon Inc., Durham, NC 27713, USAsramamoorthy@metabolon.com (P.R.); kwong@metabolon.com (K.E.W.); rsarangarajan@metabolon.com (R.S.)

**Keywords:** human liver microphysiology system, quantitative systems toxicology (QST) modeling, macromolecule, hepatotoxicity

## Abstract

Biologics address a range of unmet clinical needs, but the occurrence of biologics-induced liver injury remains a major challenge. Development of cimaglermin alfa (GGF2) was terminated due to transient elevations in serum aminotransferases and total bilirubin. Tocilizumab has been reported to induce transient aminotransferase elevations, requiring frequent monitoring. To evaluate the clinical risk of biologics-induced liver injury, a novel quantitative systems toxicology modeling platform, BIOLOGXsym™, representing relevant liver biochemistry and the mechanistic effects of biologics on liver pathophysiology, was developed in conjunction with clinically relevant data from a human biomimetic liver microphysiology system. Phenotypic and mechanistic toxicity data and metabolomics analysis from the Liver Acinus Microphysiology System showed that tocilizumab and GGF2 increased high mobility group box 1, indicating hepatic injury and stress. Tocilizumab exposure was associated with increased oxidative stress and extracellular/tissue remodeling, and GGF2 decreased bile acid secretion. BIOLOGXsym simulations, leveraging the in vivo exposure predicted by physiologically-based pharmacokinetic modeling and mechanistic toxicity data from the Liver Acinus Microphysiology System, reproduced the clinically observed liver signals of tocilizumab and GGF2, demonstrating that mechanistic toxicity data from microphysiology systems can be successfully integrated into a quantitative systems toxicology model to identify liabilities of biologics-induced liver injury and provide mechanistic insights into observed liver safety signals.

## 1. Introduction

Biologics include a wide range of products, including vaccines, blood and blood components, allergenics, somatic cells, gene therapies, tissues, and recombinant therapeutic proteins [1]. Currently, biologics represent more than 50% of the top 100, and 7 of the top 10 best-selling drugs, with monoclonal antibodies being the dominant type of biologic [2]. While biologics have shown the potential to address many unmet clinical needs, such as chronic inflammatory diseases and cancer, an increasing number of biologics-induced liver injury cases have been observed which can result in termination of clinical trials for promising treatments or liver safety warnings with recommendations to frequently monitor liver function [3]. For example, clinical development of cimaglermin alfa (GGF2), a recombinant human neuregulin-1β protein developed for heart failure therapy, was terminated due to transient elevations in serum aminotransferases and total bilirubin (biomarkers for liver injury and function, respectively) observed in Phase 1 clinical trials [4,5,6]. Tocilizumab, a human interleukin(IL)-6 receptor antagonist monoclonal antibody, initially developed for the treatment of rheumatoid arthritis, commonly induces transient aminotransferase elevations, and routine liver tests are recommended before starting tocilizumab and during treatment [3,7,8,9,10].

Prediction and prevention of biologics-induced liver injury is particularly challenging as biologics are typically developed for human targets and are not cross-reactive to other species, which limits animal-based pharmacology and toxicology studies. As such, complications may first appear during clinical trials and underlying mechanisms remain largely unknown. Standard preclinical models used for small molecule drug development are inadequate for assessing the safety of biologics [11,12]. Thus, there continues to be an unmet need for a human experimental model to assess the risk of biologics-induced liver injury in new biologics being tested for the treatment of human diseases. 

Liver damage from drugs or diseases indicates the complex relationships among multiple mechanisms and cell types. The interactions between liver hepatocytes, Kupffer cells, liver sinusoidal endothelial cells (LSECs), stellate cells, and infiltrating peripheral adaptive immune cells vacillate between regenerative and repair responses before ending with successful resolution of the insult or progression of the injury or disease. In vitro human microphysiology systems are 3D tissue/organ biomimetics that have proven their utility in predicting clinical drug exposure, efficacy, and toxicity outcomes [13,14]. The Liver Acinus MicroPhysiology System (LAMPS) is a biomimetic human liver microphysiology system model that includes multiple liver cell types: hepatocytes, LSECs, Kupffer cells, and stellate cells [15]. LAMPS recapitulates the liver acinus structure functioning in such a way as to provide a physiologically relevant platform to evaluate hepatotoxicity responses to drugs and chemicals [16]. 

In vitro mechanistic toxicity data generated in human liver microphysiology systems such as LAMPS can then be integrated with in vivo dynamic drug disposition, known biochemistry, and patient characteristics by quantitative systems toxicology (QST) modeling to predict liabilities of biologics-induced liver injury for new drug candidates or to evaluate the underlying mechanisms of clinically observed liver injury signals. QST is related to quantitative systems pharmacology (QSP) in that the two approaches apply computational modeling and multi-metric experimental methods to predict the behavior of drugs on biological systems. Both QST and QSP approaches integrate data from preclinical and clinical testing, pharmacokinetics, and mechanisms related to pharmacodynamic or toxicological effects of drugs to develop quantitative models that can be used to identify the potential therapeutic or toxicologic effects of drugs, optimize drug dosing, and support clinical trial design [17,18,19,20]. Specifically, QST modeling leverages a mathematical, mechanistic representation of physiology and of relevant biochemical pathways whereby drugs or chemicals can cause cell death and organ toxicity [21]. DILIsym^®^, a QST model of drug-induced liver injury, had been developed previously and has proven its utility in understanding and predicting liver safety liabilities and in evaluating inter-individual susceptibility and dose optimization to minimize hepatotoxicity risks for small molecules such as tolvaptan, TAK-875, and macrolide antibiotics [22,23,24,25,26]. DILIsym incorporates dynamic pharmacokinetic exposure and the ability to alter key hepatocyte pathways, informed by mechanistic studies from human-derived in vitro systems, to predict the frequency and severity of liver injury by drugs in simulated patient populations [22]. While similar approaches can be applied to predict biologics-induced liver injury, biologics present unique characteristics which distinguish them from small molecules. For example, most biologics do not penetrate into parenchymal cells, unlike small molecules, and cellular effects are mostly mediated by cell surface receptors. Additionally, monoclonal antibodies have a high degree of specificity for an antigen and often modulate immune cells or cytokines. Therefore, in the current study, BIOLOGXsym^TM^, a QST model platform focused on macromolecules, and which represents biologic-specific pathways such as receptor-mediated indirect responses and target-mediated effects, was developed to evaluate the liabilities of biologics-induced liver injury. 

The goal of this research was to investigate tocilizumab- and GGF2-induced liver injury using BIOLOGXsym combined with mechanistic toxicity testing in LAMPS. Our research demonstrated that: (1) hepatic injury and stress markers measured in LAMPS supported the idea that there is potential for two exemplar biologics, tocilizumab and GGF2, to have a negative impact on hepatocytes, consistent with clinical data; that (2) mechanistic toxicity data and metabolomics analysis from LAMPS could be utilized to characterize mechanisms of biologics-induced liver injury for tocilizumab and GGF2; and that (3) BIOLOGXsym modeling could reproduce the clinically observed hepatotoxicity of these exemplars by integrating simulated clinical exposure and mechanistic hepatotoxicity data measured in LAMPS. A larger set of biologic drugs are currently being tested to refine the predictive potential of the BIOLOGXsym platform.

## 2. Results

### 2.1. Evaluation of Tocilizumiab Effects on Phenotypic and Mechanistic Toxicity Endpoints in LAMPS Models 

Tocilizumab was tested at 1.6 µM (232 µg/mL) and 5 µM (725 µg/mL) in LAMPS models under continuous low oxygen tension (zone 3, 4–6% O_2_) media flow for 10 days. Tocilizumab concentration of 232 µg/mL is representative of the human serum C_max_ at the IV dose of 8 mg/kg [27]. Tocilizumab concentration of 725 µg/mL, which represents ~3× C_max_, was selected because it was the highest concentration achievable with available formulations. To confirm that observed tocilizumab-induced signals were a result of blocking IL-6 signaling, tocilizumab at 232 µg/mL was also tested in combination with IL-6 at 3 ng/mL, a concentration at which significant repression of cytochrome P450 (CYP)3A4 has been previously reported [27]. The significant findings are presented in Figure 1. Albumin and urea data from the individual chips are presented in the Appendix A.

Overt cytotoxicity, measured as significant lactate dehydrogenase (LDH) release, was not found in any tocilizumab treatments (Figure 1A). In addition, tocilizumab had no significant impact on hepatic function measured by albumin and urea secretion over the course of a 10-day treatment (Appendix A). However, tocilizumab treatment at 232 µg/mL produced significant increases in high mobility group box 1 protein (HMGB1) release at day 3 of treatment, and tocilizumab treatment (232 and 725 µg/mL) showed an increasing trend in HMGB1 release at day 10 of treatment (Figure 1B,C). Tocilizumab at 232 µg/mL had no effect on CYP3A4-mediated terfenadine metabolism (Figure 1D), but it increased steatosis (Figure 1E) and production of reactive oxygen species (ROS) after 10 days of treatment (Figure 1F). At the higher concentration of tocilizumab (725 µg/mL), steatosis, as measured by LipidTox, was increased (Figure 1E) and the day-10 ROS, measured by dihydroethidium, trended higher relative to control and tocilizumab at 232 µg/mL (Figure 1F). The elevated but not significant ROS increase at 725 µg/mL tocilizumab may be attributed to high variability in the measurements possibly from decreased viability (Figure 1F). Tocilizumab treatment did not significantly change the secretion of conjugated bile acid species, taurocholic acid and glycochenodeoxycholic acid (Appendix A).

IL-6 at 3 ng/mL treatment induced significant HMGB1 release on day 3 of treatment (Figure 1B), but it did not produce significant LDH release (Figure 1A). Interestingly, the combination treatment of 3 ng/mL IL-6 and 232 µg/mL tocilizumab significantly increased LDH release at day 6 of treatment (Figure 1A). However, IL-6 or the combination tocilizumab and IL-6 treatments had no effects on the albumin and urea measurements (Appendix A). IL-6 significantly decreased fexofenadine formation from CYP3A4-mediated terfenadine metabolism (Figure 1D), consistent with a previous report [27]. The effect on CYP3A4 activity was rescued when tocilizumab was co-treated with IL-6 (Figure 1D).

Overall, injury markers measured in LAMPS provided some evidence of hepatic stress and injury, generally indicating the potential for negative impacts of tocilizumab on hepatocytes [15,16,28]. HMGB1 was shown to be a more sensitive indicator of injury compared with LDH, potentially due to differences in release mechanisms. HMGB1 is actively secreted or passively released from damaged hepatocytes, Kupffer cells and LSECs, whereas LDH is released from hepatocytes undergoing damage or death [29]. In addition, targeted mechanistic endpoints and metabolomics analysis demonstrated that tocilizumab induced steatosis and oxidative stress. These mechanistic outputs were subsequently employed to determine the hepatotoxicity mechanisms and parameters of tocilizumab in BIOLOGXsym simulations, as described in the following sections.

### 2.2. Metabolomics Analysis of Tocilizumab Effects in LAMPS Models

Metabolomics profiling was performed by Metabolon, Inc. Durham, NC, USA, using the untargeted Global Discovery Panel. Tocilizumab treatment at 1.6 µM (232 µg/mL) and 5 µM (725 µg/mL) in the LAMPS model under continuous flow for 10 days was associated with persistent and significant alterations in several metabolic markers of oxidative stress (Figure 2). Methionine, an amino acid with potent antioxidative properties [30], was significantly decreased following tocilizumab treatment at both concentrations on day 6 and day 10 compared with their respective vehicle controls. Similarly, several metabolites in the oxidative stress pathway, including methionine sulfone, methionine sulfoxide [31], cysteine (an amino acid with antioxidant properties), and cystine were also lower with treatment. In contrast, taurine, a natural modulator of the antioxidant defense network, demonstrated relatively higher levels with the lower dose of tocilizumab (232 µg/mL), notably significant at day 10 (Figure 2).

Although glutathione (GSH), a tripeptide antioxidant, was not detected in the dataset, several metabolites associated with GSH biosynthesis and turnover such as cysteine-glutathione disulfide, 5-oxoproline, and gamma glutamyl amino acids were decreased compared with the vehicle control (Figure 2). Another key player in hepatic redox homeostasis is nicotinamide adenine dinucleotide (NAD+), a hydride group acceptor in the oxidation of carbohydrates, amino acids, and fats. Following 232 µg/mL tocilizumab treatment, significantly higher levels of quinolinate and nicotinate were observed, suggesting an increase in NAD+ synthesis. However, treatment at the 725 µg/mL tocilizumab dose was not associated with any changes in the levels of quinolinate and nicotinate. NAD+ is inherently an intracellular biochemical that is rapidly utilized by a range of enzymatic reactions, which explains the low levels of NAD+ measured here in the effluents [32]. 

Of interest in the study are the metabolomics outputs that support the relationship between IL-6 and its influence on liver inflammation and fibrosis. Metabolomic profiling of the LAMPS effluents demonstrated an increase in a group of metabolites related to protein turnover and extracellular matrix remodeling. There is also evidence suggesting a role for IL-6 in liver inflammation and fibrosis, and the metabolomic profile observed in this study is suggestive of altered liver vascular homeostasis, a typical manifestation of liver pathologies [33]. Notable changes in metabolites associated with tissue and extracellular remodeling have been graphically presented and discussed in Appendix A.

### 2.3. Evaluation of GGF2 Effects on Phenotypic and Mechanistic Toxicity Endpoints in LAMPS Models 

GGF2 at 10, 100, and 382 ng/mL were tested in the LAMPS model, with significant findings presented in Figure 3. Albumin and urea data from the individual chips are found in the Appendix A.

No overt toxicity was identified by LDH release at any GGF2 concentration tested (Figure 3A), but a significant level of HMGB1 release was found at 382 ng/mL GGF2 on day 3 of treatment (Figure 3B). GGF2 had no effects on hepatic function measured by albumin and urea secretion (Appendix A).

Steatosis was significantly elevated at 10 and 382 ng/mL GGF2, but only trended higher (not statistically significant) relative to control at 100 ng/mL GGF2 (Figure 3C). The lack of significant steatosis increase at 100 ng/mL GGF2 can be attributed to high variability in the data (Figure 3C). No effect of GGF2 on ROS production was found in these studies (Figure 3D). GGF2 treatment resulted in a significant decrease in the secretion of conjugated bile acid species, taurocholic acid and glycochenodeoxycholic acid (Figure 3E,F) [4].

Overall, GGF2 increased HMGB1 in LAMPS, providing evidence for hepatocyte stress and the potential induction of an inflammatory response. In addition, the targeted mechanistic endpoints demonstrated that GGF2 reduced bile acid secretion; these were subsequently employed to determine the hepatotoxicity mechanisms and parameters of GGF2 in BIOLOGXsym simulations as described in following sections. While LAMPS experiments showed increased steatosis after 10 days, no changes in ROS (lipotoxicity) were observed; hence, the steatosis-induced lipotoxicity pathway within BIOLOGXsym was not parameterized for GGF2-related effects. 

### 2.4. Determination of Tocilizumab Mechanistic Toxicity Parameters

An in-vitro-like setup within BIOLOGXsym mimicking the LAMPS experiments was able to reproduce steatosis and ROS signals (% of control) observed in tocilizumab (232 and 725 µg/mL)-treated LAMPS by optimizing the relevant parameters (Figure 4; Appendix A). Steady-state hepatic interstitial concentrations of tocilizumab corresponding to the 232 and 725 µg/mL LAMPS dosing concentrations were predicted by a physiologically based pharmacokinetic (PBPK) model for tocilizumab (developed as described in the Appendix A) to be 52.4 and 166.3 µg/mL, respectively (Figure 4A,D), which were assumed to drive tocilizumab mechanisms of toxicity (i.e., induction of hepatic steatosis and ROS). Empirical tocilizumab-mediated inhibition of very-low-density lipoprotein (VLDL)-triglyceride release reasonably recapitulated the steatosis (% of control) observed on day 10 in LAMPS (Figure 4B,E). Under these in-vitro-like conditions, the BIOLOGXsym model predicted that the extent of steatosis (i.e., 203 and 204% of control for 232 and 725 µg/mL tocilizumab, respectively) on day 10 was not sufficient to explain the accumulation of ROS observed in LAMPS on that experimental day. Additional ROS buildup, represented by a saturable tocilizumab concentration-dependent ROS production rate in BIOLOGXsym, was able to reproduce the ROS signal on day 10 of the LAMPS experiments for both tested tocilizumab concentrations (i.e., 220 and 230% of control, respectively; Figure 4C,F).

In addition, the effects of tocilizumab on IL-6 receptor signaling and selected downstream effects which were potentially relevant to tocilizumab-mediated hepatotoxicity (i.e., hepatocyte regeneration, macrophage recruitment, suppression of CYP expression) were represented (Appendix A). An in-vitro-like setup within BIOLOGXsym, mimicking the LAMPS experiments, was able to reproduce a dose-dependent reduction in CYP3A4 activity in response to IL-6 increases, consistent with vLAMPS experiments and [27]. This reduction was reversed by tocilizumab (Appendix A). 

### 2.5. Simulations of Tocilizumab-Mediated Hepatotoxicity in BIOLOGXsym 

Simulations were performed with a SimCohorts^TM^ (n = 4) with elevated IL-6 levels, which serves as a targeted proxy for the pro-inflammatory rheumatoid arthritis disease setting. Tocilizumab has been shown to change the expression level of CYP enzymes and may exacerbate the hepatotoxicity responses to drugs whose hepatotoxic effects are driven by CYP-mediated metabolite formation, such as N-acetyl-*p*-benzoquinone (NAPQI) production from acetaminophen. To fully examine the potential hepatotoxic effect of tocilizumab, simulations were performed with tocilizumab alone, repeated therapeutic dosing of acetaminophen alone (1 g four times a day (4 g/day) for 12 weeks), and tocilizumab co-administered with acetaminophen. Proof-of-concept simulations of biologics-induced liver injury responses for tocilizumab in BIOLOGXsym were made by integrating tocilizumab-induced ROS and steatosis, tocilizumab effects on IL-6 receptor signaling and selected downstream effects which were potentially relevant to tocilizumab-mediated hepatotoxicity (i.e., hepatocyte regeneration, macrophage recruitment, suppression of CYP expression), and by the hepatic exposure predicted by GastroPlus^®^ PBPK modeling for a clinically reported tocilizumab protocol (i.e., 8 mg/kg IV tocilizumab given every four weeks for 12 weeks). 

Peak alanine aminotransferase (ALT) responses in these simulations are summarized in Figure 5. When tocilizumab was administered alone, ALT elevations > 1× the upper limit of normal (ULN, defined as 40 U/L for ALT) were predicted for all individuals, and one individual showed ALT > 3× ULN (Figure 5A). This response was consistent with the frequent ALT elevations > 1× ULN (33.8%) and less frequent ALT elevations of 3–5× ULN (3%) and >5× ULN (2%) reported in 288 patients treated with tocilizumab [7]. In the subsequent analysis, mechanisms underlying increased ALT levels by tocilizumab treatment were investigated by simulating two scenarios: tocilizumab-mediated inhibition of explicitly modeled IL-6 effects only (changes in CYP activity, hepatocyte regeneration, macrophage recruitment, and IL-6 cross-signaling effects) vs. tocilizumab-mediated steatosis and ROS elevation only. Inhibiting explicit IL-6 activity alone did not predict any ALT elevations, while tocilizumab-mediated steatosis and ROS elevations alone predicted ALT elevations for all individuals. These results suggest that ROS induction is the main mechanism underlying simulated ALT elevations because simulations of in-vitro-like conditions showed that the extent of steatosis observed in LAMPS was not sufficient to induce lipotoxicity (i.e., oxidative stress) as described in the prior section. Inhibiting IL-6 activity was found to slow the recovery of viable hepatocytes post-injury due to the mitigation of IL-6-mediated hepatocyte regeneration (Appendix A).

For individuals administered acetaminophen alone, ALT increases above 3× ULN were seen in the two individuals with increased susceptibility to ROS-driven toxicity. When tocilizumab and acetaminophen were co-administered, all individuals showed mild to severe increases in ALT (i.e., ALT elevations > 1, >3, or >5× ULN). The two individuals with increased susceptibility to ROS-driven toxicity experienced ALT > 5× ULN, substantially higher than the ALT response seen with acetaminophen alone or tocilizumab alone. In the subsequent analysis, mechanisms underlying increased ALT levels during co-administration of tocilizumab and acetaminophen were investigated by simulating two scenarios: co-administration including tocilizumab-mediated inhibition of explicitly modeled IL-6 signaling only (changes in CYP activity, hepatocyte regeneration, macrophage recruitment, and IL-6 cross-signaling effects) and co-administration including tocilizumab-mediated steatosis and ROS elevations only. Notably, changes in CYP activity that would affect acetaminophen-driven hepatotoxicity were not included in the second scenario. In both scenarios, ALT increases higher than those seen with acetaminophen alone in the high-ROS susceptibility individuals were predicted. These results suggest that the effects of tocilizumab on both the explicitly modeled IL-6 effects and the tocilizumab-mediated ROS elevations contribute to increased liver injury signals during co-administration of tocilizumab with acetaminophen.

### 2.6. Determination of GGF2 Mechanistic Toxicity Parameters 

GGF2 mechanistic toxicity pathways were parameterized using a combination of in vitro data from the literature and data from LAMPS experiments reported herein. Model parameters representing alterations of regulatory pathways for hepatic enzyme and transporter expression related to bile acid and bilirubin disposition were initially estimated based on published transcriptional data on GGF2-treated primary human hepatocytes [4]. Because the magnitude of transcriptional changes does not always translate to the magnitude of functional changes, glycochenodeoxycholic acid and taurocholic acid secretion data obtained from GGF2-treated LAMPS were utilized to further optimize parameter values (Appendix A). An in-vitro-like setup within BIOLOGXsym mimicking the LAMPS experiments was able to reasonably reproduce the decreased bile acid secretion (basolateral and biliary efflux processes combined; % of control) observed in GGF2 (10, 100, and 382 ng/mL)-treated LAMPS (Figure 6). In this in-vitro-like setup, steady-state hepatic interstitial concentrations of GGF2 corresponding to the 10, 100, and 382 ng/mL LAMPS dosing concentrations were predicted by a PBPK model for GGF2 (developed as described in the Appendix A) to be 33.6, 319.6, and 1066.4 ng/mL, respectively, and were assumed to serve as the driving force for toxicity.

### 2.7. Simulations of GGF2-Mediated Hepatotoxicity in BIOLOGXsym

Proof-of-concept simulations of biologics-induced liver injury responses for GGF2 in BIOLOGXsym were made by integrating the inhibitory effects of GGF2 on the disposition of bile acids and bilirubin, as well as hepatic exposure predicted by GastroPlus PBPK modeling for a clinically reported GGF2 protocol (a single IV dose of 1.5 mg/kg) [6]. Simulated plasma ALT and total bilirubin levels in BIOLOGXsym were compared with clinical data, in which 2 out of 40 treated subjects showed concomitant elevations in ALT and total bilirubin [5]. Initial simulations in the baseline human using GGF2 toxicity parameters informed by published transcriptional data on bile acid- and bilirubin-related enzymes or transporters and mechanistic LAMPS data predicted minimal changes in plasma ALT and an increase in plasma total bilirubin, underpredicting clinically observed ALT responses. As such, a sensitivity analysis was performed for the impact of GGF2 on various bilirubin- and bile-acid-related mechanisms in vivo to account for uncertainty around transcriptional data and translation of LAMPS bile secretion data for use in in vivo predictions. In addition, simulations were conducted in a SimCohorts consisting of 16 simulated individuals that represent varying levels of patient susceptibility because only a select number of subjects treated with a single IV dose of GGF2 at 0.38 mg/kg or 1.5 mg/kg experienced liver injury biomarker elevations in clinical trials, indicating that observed clinical signals were attributed in part to patient-specific susceptibility factors.

The sensitivity analysis indicated that GGF2 inhibitory effects on basolateral efflux could lead to clinically relevant ALT elevations, while only minimally impacting total bilirubin levels (Figure 7). Inhibitory effects of GGF2 on (un)conjugated bilirubin uptake as well as stimulatory effects on the basolateral efflux of conjugated bilirubin were shown to substantially elevate total bilirubin levels in the simulations. These simulations suggest that the predicted ALT elevations may be attributed to a GGF2-induced accumulation of centrilobular amidated chenodeoxycholic species, particularly when basolateral transport of bile acids is reduced substantially. While GGF2 interactions with other bile acid pathways (e.g., biliary efflux, uptake, amidation) reproduced neither ALT nor total bilirubin elevations in the baseline human, inhibition of hepatocellular bile acid uptake or biliary bile acid excretion was predicted to slightly reduce hepatic adenosine triphosphate (ATP) levels, indicating sub-lethal hepatotoxicity responses. Based on sensitivity analysis results, GGF2 toxicity parameters were further optimized to recapitulate clinical data (Appendix A). Final simulations incorporating inter-individual variability based on optimized GGF2 toxicity parameters reproduced the range, timing, and inter-individual variability of GGF2-related clinically observed liver signals (Figure 8).

## 3. Discussion

Microphysiology systems have gained traction in toxicity testing because they overcome limitations of animal pharmacology and toxicological studies (e.g., species differences) and two-dimensional cell culture systems (e.g., lack of non-parenchymal cells and organ architecture). Furthermore, experimental data obtained from microphysiology systems can be integrated into a database such as BioSystics-Analytical Platform™ (BioSystics-AP™, see Section 4) which has been harnessed to access, analyze, manage, share and computationally model data from microphysiology systems (see Section 4) [28,34]. For example, a QSP platform merging computational and experimental protocols of liver disease in microphysiology systems has been developed and has proven successful. The computational analysis of a patient-derived hepatic RNA sequencing dataset predicted that a histone deacetylase inhibitor, vorinostat, would reduce liver fibrosis and inflammation in non-alcoholic fatty liver disease (NAFLD) patients, and has been successfully reproduced in the LAMPS model of progressive NAFLD [17]. In recent years, QST modeling has demonstrated its utility in predicting organ toxicity, and the current study integrated microphysiology systems and QST models into one platform technology. 

Mitochondrial toxicity, induction of ROS, inhibition of bile salt export protein, and endoplasmic reticulum stress are mechanisms which have been associated with clinical liver injury [35,36,37]. In hepatocytes, ROS can be a product of mitochondrial electron transport chain during oxidative phosphorylation, lipogenesis, steatosis, or drug metabolism [38,39]. Excess ROS can lead to oxidative stress, which damages hepatocellular membranes, DNA, and proteins, leading to cellular dysfunction, apoptosis, necrosis, and organ toxicity [37,40]. In the current study, tocilizumab induced hepatocyte steatosis and ROS in LAMPS. Metabolomics analysis of the effluent from the LAMPS model system exposed to tocilizumab showed altered signatures of redox homeostasis, thus, supporting the cellular imaging results. In the liver, both enzymatic and non-enzymatic antioxidant systems serve as a defense against harmful reactive species. Sulfur-containing amino acids (methionine and cysteine) comprise a crucial component of antioxidative defense mechanism in the liver [41]. In this study, methionine and oxidized derivatives such as methionine sulfone and methionine sulfoxide were lower in tocilizumab-treated LAMPS efflux media, which is suggestive of potential ROS neutralization pathway activation [30]. Methionine is also a precursor for cysteine, an amino acid with moderate antioxidative properties that donates a hydrogen from its thiol group to become oxidized to its corresponding derivative cystine (cysteine/cystine redox couple) [42]. In this study, cystine levels were consistently decreased in the efflux from tocilizumab-treated LAMPS systems, suggesting alterations in cysteine/cystine cycling within liver cells as a potential mechanism to cope with tocilizumab-treatment-induced ROS. Cysteine also plays an important role in redox homeostasis by serving as a component of GSH [43]. Several metabolites involved in its biosynthesis/turnover such as cysteine-glutathione disulfide, 5-oxoproline, and gamma glutamyl amino acids (markers of glutathione cycle activity [44]) have demonstrated decreased levels compared with the vehicle-treated groups. The observed changes suggest glutathione depletion and/or its related metabolites within the tissue or, alternatively, their high consumption rates within the cells. Collectively, metabolomics analysis of efflux media provides a snapshot of the elevated oxidative stress environment and reveals the altered levels of many endogenous antioxidant metabolites within the LAMPS model system following tocilizumab treatment.

Previous hypotheses for clinically observed ALT elevations include loss of the hepatoprotective effects of IL-6 [45]. IL-6 stimulates hepatocyte regeneration pathways which, when blocked by tocilizumab, could leave the liver susceptible to injury. To test this hypothesis, selected downstream effects of IL-6 receptor that might potentially be relevant to biologics-induced liver injury pathways (i.e., hepatocyte regeneration [46,47,48], macrophage recruitment [49,50,51,52], and suppression of CYP expression [53,54,55,56,57,58]) were represented within BIOLOGXsym. Blockade of IL-6 signaling by tocilizumab would reduce the regenerative capacity of the liver, decrease the macrophage recruitment, and reverse the IL-6 receptor-mediated suppression of hepatic CYP isozymes. 

Simulations of a clinical protocol of tocilizumab in BIOLOGXsym leveraging the mechanistic toxicity data from LAMPS (i.e., induction of steatosis and ROS) and blockade of IL-6 signaling reproduced clinically observed and modest ALT elevations. Subsequent mechanism analysis suggested that modest ALT elevations could be attributed to the ROS induction observed in the LAMPS studies rather than the blockade of IL-6 signaling. While the effects of tocilizumab on IL-6 signaling did not contribute to elevations of ALT, they led to the slower recovery of viable hepatocytes. Overall, these results suggest that tocilizumab-mediated ROS induction may lead to modest liver injury in a subset of susceptible patients, while tocilizumab effects on IL-6 signaling may delay the recovery post injury. To our knowledge, this is the first report to show tocilizumab-mediated ROS induction and its contribution to clinically observed modest ALT signals. Of note, the underlying mechanisms of tocilizumab-mediated ROS induction still remain unknown. While tocilizumab-mediated ROS induction was not impacted by the co-incubation of IL-6 in LAMPS assays and the tocilizumab-soluble IL-6 receptor complex has been reported to be incapable of inducing downstream signaling [59,60], we cannot rule out IL-6 receptor blockade as having indirect downstream effects leading to increases in ROS, as IL-6 has broad effects on multiple cell types, including hepatocytes. 

The enhanced ALT elevations during simulated tocilizumab and acetaminophen co-administration emphasize the potential for drug–drug interactions at both pharmacokinetic and toxicological pathways. Acetaminophen has been shown to induce significant liver injury after overdose and mild ALT elevations in a subset of patients administered repeat therapeutic doses [61,62]. The underlying mechanism of acetaminophen-mediated hepatotoxicity has been extensively studied in humans and animals; NAPQI, a reactive metabolite of acetaminophen which is formed by CYP2E1 and CYP3A4, binds to intracellular macromolecules, induces oxidative stress, and lead to cell death, when detoxification by glutathione is overwhelmed [63]. Simulation of a combination treatment of tocilizumab and acetaminophen predicted enhanced liver injury compared to each drug administered alone. Mechanism analyses suggested that tocilizumab-mediated ROS induction (which was added to NAPQI-mediated ROS induction) and reduced IL-6 signaling (i.e., increased expression of CYP enzymes that led to increased concentrations of NAPQI) contributed to enhanced ALT elevations in simulations of tocilizumab combined with acetaminophen. Although clinical data on toxicological interactions of tocilizumab and acetaminophen are scarce, it has been reported that blockade of IL-6 trans-signaling increased the serum ALT and aggravated liver injury in mice treated with acetaminophen, consistent with simulation results [57]. Overall, these results suggest that multiple toxicity mechanisms could independently account for hepatotoxicity and contribute to the relatively frequent observation of mild ALT elevations in tocilizumab-treated patients. They also demonstrate that QST modeling can evaluate drug–drug interactions for toxicological responses by integrating relevant mechanisms for each drug. 

The BIOLOGXsym simulations of GGF2 described herein incorporated a representation of GGF2 effects on transcriptional regulation of bile acid and bilirubin disposition and were thereby able to reasonably reproduce liver injury biomarker elevations observed in subjects enrolled in Phase 1 clinical trials [4,5,6]. This was achieved by taking into account population variability in susceptibility factors and uncertainty in GGF2-related parameter values, demonstrating the importance of including this type of variation in the current model and simulations. GGF2 parameters were informed by transcriptional data [4] and optimized with bile acid secretion data in LAMPS. Further parameter optimization was necessary to explain clinically observed plasma ALT elevations, indicating that scaling factors or an in-vitro-to-in-vivo extrapolation model may be needed to translate LAMPS data to BIOLOGXsym inputs for in vivo predictions [64,65,66]. Additional applications of both the LAMPS and BIOLOGXsym platforms with multiple exemplar drugs would be necessary to potentially establish such scaling factors. 

Additional insights into the functional effects of GGF2 have been derived from data on bile acid modulation in sandwich-cultured human hepatocytes, showing that GGF2 may diminish the biliary clearance and total accumulation of a probe bile acid substrate (d_8_- taurocholic acid) [4]. Those findings indicate that vectorial bile acid transport from medium into the bile pocket was reduced by GGF2 treatment. On the other hand, GGF2 did not reduce d_8_- taurocholic acid’s biliary excretion index in sandwich-cultured human hepatocytes, which is a measure of bile acid transport from within the hepatocellular space into bile. This suggests that GGF2 may not functionally impair bile salt export pump (BSEP)-mediated efflux of bile acids, and that GGF2-mediated reduction of biliary bile acid clearance may be the result of decreased bile acid uptake. Measurements of bile acid secretion in the LAMPS experiments represent a combination of basolateral and biliary efflux processes, implying that the measured decreases in glycochenodeoxycholic acid and taurocholic acid secretion upon GGF2 treatment could be a consequence of either pathway being impaired. However, if BSEP function is minimally altered by GGF2 treatment, as suggested by the biliary excretion index of d_8_-taurocholic acid upon GGF2 treatment in sandwich-cultured human hepatocytes, it is plausible that reduction of basolateral bile acid transport predominates in order for bile acid secretion in the LAMPS experiments to be decreased. Simulations in BIOLOGXsym show that substantial downregulation of this pathway can indeed reproduce the LAMPS bile acid secretion data and furthermore explain the clinically observed elevations in plasma ALT.

Current BIOLOGXsym simulations suggest that the hepatocellular injury corresponding to the simulated levels of clinically observed ALT elevations is not sufficient to explain the clinically observed total bilirubin elevations. Transcriptional GGF2 data from primary human hepatocytes on bilirubin disposition were utilized in the current simulations to explain elevations in plasma total bilirubin [4]. This is in agreement with a previous simulation analysis of plasma ALT and total bilirubin levels using DILIsym, which concluded that liver dysfunction arising from GGF2 treatment was insufficient to significantly increase total bilirubin [5]. Furthermore, GGF2-mediated reduction in biliary bile acid clearance, as shown in sandwich-cultured human hepatocytes, could lead to a decrease in bile-acid-dependent bile flow, which can also impact the biliary excretion of bilirubin and contribute to elevated bilirubin levels. This is a component not currently measured within the BIOLOGXsym model for GGF2.

While the GGF2 simulations were informed by published transcriptional data, future studies combining the LAMPS and BIOLOGXsym platforms would ideally provide those additional types of mechanistic data without relying on the literature. Suitable experimental methodology would need to be applied that can distinguish transcriptional, or preferably translational, alterations from the different cell types used in the LAMPS model (i.e., hepatocytes, stellate cells, Kupffer cells, LSECs). Cell-type-specific alterations in key metabolic pathways are critical to understand hepatotoxicity mechanisms and generating insightful in vivo predictions about hepatic phenotypes and eventually pathological outcomes, thus, integrating metabolomics analysis will greatly enhance the capabilities of the LAMPS and BIOLOGXsym platforms. In addition, simulated populations that represent variability in mechanistic pathways will need to be developed to fully represent inter-individual variabilities in susceptibility to hepatotoxicity. Adaptive immune responses, which have been proposed as key contributors underlying biologics-mediated hepatotoxicity, especially of immune checkpoint inhibitors, will need to be represented to evaluate hepatotoxicity responses via this mechanism. While further efforts are being made to address these limitations, the current study has demonstrated a proof of concept that mechanistic toxicity data from microphysiology systems can be successfully integrated into a QST model to evaluate biologics-induced liver injury liabilities and provide mechanistic explanation for observed liver safety signals.

## 4. Materials and Methods

### 4.1. The Liver Acinus Microphysiology System (LAMPS)

We employed our previously published human LAMPS model to study the effects of tocilizumab and GGF2 on LAMPS health and function [16,67]. A key advantage to the use of the LAMPS model is having the capability to test drugs in a liver zone 1 or a zone 3 oxygenated device to take advantage of the differences in function and metabolism between the liver zones which places hepatocytes surrounding the central vein (zone 3) in the low oxygen gradient which is most often the initiation site for liver injury and disease [68]. In these studies, we focused on tocilizumab and GGF2 treatment in a LAMPS model with a flow rate set at 5 µL/h to create the hepatic zone 3 oxygen tension [16]. The 120 µL/day media volume collected was sufficient to measure albumin, urea, LDH, HMGB1, and liquid chromatography mass spectrometry (LC MS/MS) for conjugated bile acids and CYP3A4-mediated terfenadine metabolism.

### 4.2. LAMPS Model Assembly and Maintenance

A schematic of the LAMPS model is provided in Appendix A. The LAMPS setup was carried out as previously described [15,16] with the added modification to replace the EA.hy926-transformed endothelial cells with primary LSECs obtained from LifeNet Health. The LAMPS model was constructed in a single chamber commercial microfluidic device, ParVivo™ (SCC-001) available from Nortis, Inc. Seattle, WA, USA. In all steps involving injection of media and/or cell suspensions into LAMPS devices, an amount of 100–150 μL per device was used to ensure complete filling of fluidic pathways, chamber and bubble traps. The percentages of hepatocytes, THP-1 Kupffer-like cells, LSEC, and LX-2 stellate cells are consistent with the scaling used in our previously published models [15,16,67,68] and presented as Table 1. For the studies described here, LAMPS models were set up and maintained for 10 days under flow.

### 4.3. Normal-Fasting Media

The media used in the studies is developed around a custom Williams E media that allows us to adjust the glucose, insulin, and glucagon to reflect normal fasting serum levels for glucose, insulin and glucagon. The final media consisted of 5.5 mM glucose (Millipore, Burlington, MA, USA), 1% FBS (Corning, Corning, NY, USA), 0.125 g/mL bovine serum albumin (Sigma, St. Louis, MO, USA), 0.625 mg/mL human transferrin, 0.625 μg/mL selenous acid, 0.535 mg/mL linoleic acid (Sigma), 100 nM dexamethasone (ThermoFisher, Waltham, MA, USA), 2 mM glutamax, 15 mM HEPES (ThermoFisher), 100 U/100 μg/mL pen/strep (HyClone Labs, Logan, UT, USA), 10 pM insulin (ThermoFisher), and 100 pM glucagon (Sigma) [20].

### 4.4. Drug Preparation and Treatment 

Tocilizumab was purchased from Selleck Chemicals (Houston, TX, USA) (A2012) and GGF2 active protein was purchased from MyBioSource (San Diego, CA, USA) (MBS553231). Tocilizumab was received as a 5 mg/mL solution. GGF2 was prepared at 1 mg/mL in PBS. Tocilizumab and GGF2 stock drug preparations were stored at 4 °C and −20 °C, respectively. Working concentrations of tocilizumab at 232 and 725 µg/mL were prepared using the normal-fasting media. Due to the large dilutional effect to prepare tocilizumab at 232 and 725 µg/mL (1.6, 5 µM), the media supplements listed in the normal-fasting media preparation methods were added to adjust the concentrations. No adjustments to the normal-fasting media supplements were necessary for GGF2 concentrations at 10, 100, and 382 ng/mL. 

### 4.5. Device Fixation Protocol 

At the termination of the studies, treatment media was aspirated from the devices. The devices were rinsed one time in PBS which was replaced with 4% paraformaldehyde solution for 20 min. The formalin fixative was aspirated and replaced with 0.5% Triton X-100 for 15 min. The Triton X-100 media was removed, the device was washed twice in PBS and then treated to LipidTox dye staining before image collection and analysis (see Appendix A). 

### 4.6. Assessment of Drug Effects on Outcome Measurements in LAMPS 

Effects of tocilizumab, IL-6, and GGF2 on hepatocyte steatosis, secretome (albumin, urea, LDH, various human cytokines, bile efflux), and ROS were measured in LAMPS as described in the Appendix A. Metabolomic profiling was performed using the effluent of LAMPS treated with tocilizumab as described in Appendix A.

### 4.7. Statistical Analysis 

Statistical comparisons between specific treatment and control conditions were made by performing One-Way ANOVA analysis without assuming equal SDs (i.e., Brown–Forsythe and Welch ANOVA tests) using GraphPad Prism version 9.4.0 (San Diego, CA, USA) at a significance level α of 0.05. Follow-up comparisons of the mean of each treatment condition with the control condition were performed using Dunnett’s T3 multiple comparisons test. In the case of statistical comparisons between only one treatment condition and the control condition, *t* tests without assuming equal SDs were performed, at an α of 0.05. 

### 4.8. BioSystics-Analytics Platform (BioSystics-AP)

All data generated in these studies are available through the BioSystics-AP web site at (biosystics-ap.com) for review and download by any registered user. The BioSystics-AP was developed with support by NCATS to be a platform to store and access data and metadata and to manage, analyze and model data from microphysiology systems with links to integrated federated databases on one site where all can be accessed [28,34]. 

### 4.9. Development of BIOLOGXsym 

BIOLOGXsym, a QST model to predict biologics-induced liver injury liabilities, was developed in MATLAB 2021a (MathWorks, Natick, MA, USA). BIOLOGXsym is a deterministic model which is composed of ordinary differential equations and algebraic expressions that represent essential biochemical and physiological components related to liver health and toxicity. BIOLOGXsym integrates various sub-models that represent physiological processes involved in biologics-induced liver injury into a single simulation (e.g., hepatocyte life cycle, oxidative stress, mitochondrial dysfunction and toxicity, bile acid disposition, and biomarker release), generating continuous, time-dependent simulation outputs such as hepatic biomarkers and measures of hepatic stress and injury (e.g., liver ATP, fraction of viable hepatocytes). This approach is similar to the previously developed QST model of drug-induced liver injury focused on small molecules [22,23,24,25,26]. However, BIOLOGXsym represents pathways and mechanisms unique to biologics such as cell-surface-receptor-mediated indirect responses and target-mediated effects. More information about BIOLOGXsym can be found in Appendix A.

### 4.10. Development of Physiologically-Based Pharmacokinetic (PBPK) Models of Tocilizumab and GGF2 

PBPK models for tocilizumab and GGF2 were developed within GastroPlus^®^ 9.8 (Simulations Plus, Lancaster, CA, USA) to predict the hepatic drug exposure in humans. Details regarding the PBPK modeling are provided in the Appendix A. Tocilizumab and GGF2 parameters employed in the PBPK models (obtained from experimental data or optimized to clinical pharmacokinetic data) are presented in Appendix A. Comparison of simulated vs. observed plasma concentration–time profiles and pharmacokinetic parameters are presented in Appendix A. 

### 4.11. Hepatotoxicity Mechanisms and Parameters of Tocilizumab 

To represent tocilizumab in BIOLOGXsym, mechanistic models of the major on-target pathways affecting hepatocytes were developed. IL-6 signaling through both soluble and membrane bound IL-6 receptors were represented. For soluble IL-6 receptor, IL-6 binding soluble IL-6 receptor forming an IL-6-soluble–IL-6 receptor complex was explicitly represented. In this pathway, tocilizumab modulates the concentration of free soluble IL-6 receptors, thereby reducing the concentration of the IL6-soluble–IL-6 receptor complex formed. For membrane-bound IL-6 receptor, an implicit representation was developed where an ‘effective’ IL-6 pool represents the concentration of free IL-6 needed to produce a given signal through the membrane-bound receptor. In this case, tocilizumab administration mimics the binding of membrane-bound IL-6 receptor by reducing this effective IL-6 concentration.

Key downstream effects were included that could impact hepatocyte health, including changes in CYP activity, hepatocyte regeneration, and macrophage recruitment to the liver. Timing and magnitude of response in relation to membrane-bound/soluble IL-6 receptor signaling were parameterized based on data available from LAMPS and the literature [27,56].

Steatosis and ROS signals (% of control) obtained from tocilizumab (232 µg/mL)-treated LAMPS were used for parameterization of tocilizumab effects on the induction of steatosis and ROS, respectively, in BIOLOGXsym. An increase in steatosis was represented by hypothetical tocilizumab-mediated inhibition of VLDL-triglyceride release, which would lead to elevations in hepatic triglycerides, and thus, fatty liver. While an increase in steatosis can contribute to ROS signals, additional ROS generation is represented in BIOLOGXsym by a non-linear RNS/ROS production rate dependent on the concentration of tocilizumab. Since tocilizumab concentrations in the hepatic interstitium were assumed to drive the intrinsic toxicity signals, PBPK modeling was used to simulate hepatic interstitial concentration of tocilizumab. To reasonably predict the hepatic interstitial concentrations of tocilizumab in the LAMPS experiments, drug exposure was simulated in GastroPlus for an in-vitro-like environment. Constant IV infusions of tocilizumab were optimized to obtain steady-state plasma concentrations of tocilizumab at 232 and 725 µg/mL, as a reasonable representation of the dosing and media concentration of tocilizumab in LAMPS. The corresponding, predicted steady-state hepatic interstitial concentrations in LAMPS were subsequently used to parameterize tocilizumab’s intrinsic toxicity pathways.

### 4.12. Setup of Tocilizumab-Mediated Hepatotoxicity Simulations in BIOLOGXsym

A SimCohorts (n = 4) consisting of two acetaminophen (1 g four times a day (4 g/day) for 12 weeks) hepatotoxicity responders (ALT > 3× ULN) and two non-responders (no ALT elevations) was used to assess tocilizumab and acetaminophen interactions when IL-6 levels are elevated in subjects. Tocilizumab (8 mg/kg, every 4 weeks (q4wk) for 12 weeks) was administered alone or in combination with acetaminophen (1 g four times a day (4 g/day) for 12 weeks). DILIsym v8A was used to simulate NAPQI formation from acetaminophen [71,72,73] after incorporating IL-6- and tocilizumab-mediated effects on CYPs. Simulated NAPQI profiles and toxicity associated with this reactive metabolite of acetaminophen were implemented in BIOLOGXsym simulations with and without tocilizumab. Furthermore, a sensitivity analysis was performed to evaluate the individual roles of tocilizumab’s intrinsic toxicity (i.e., ROS generation) and membrane-bound IL-6-receptor-mediated effects (i.e., CYP alterations, hepatocyte regeneration, macrophage recruitment) when co-administered with acetaminophen.

### 4.13. Hepatotoxicity Mechanisms and Parameters of GGF2

Mechanistic in vitro data from LAMPS experiments and the literature were used to derive mechanistic toxicity parameters within BIOLOGXsym for GGF2. Potential effects of GGF2 on liver pathophysiology include alterations of regulatory pathways for hepatic enzyme and transporter expression [4]. Transcriptional profiling data on the impact of GGF2 on bile acid disposition (i.e., decreases in mRNA of sodium/taurocholate cotransporting polypeptide (NTCP/SLC10A1)), multidrug-resistance-associated protein 4 (MRP4/ABCC4), bile salt export pump (BSEP/ABCB11), bile acid-coenzyme A:amino acid N-acyltransferase (BAAT), fatty acid transport protein 5 (FATP5/SLC27A5/BACS) and bilirubin transport (decrease in organic anion transporting polypeptide 1B3 (OATP1B3) mRNA, and increase in MRP3 mRNA) were furthermore utilized to inform the parameterization of GGF2 effects on these pathways [4]. Alterations in glycochenodeoxycholic acid and taurocholic acid secretion observed 4–6 days after GGF2 (10–382 ng/mL) treatment using the LAMPS model, were used to optimize parameter values in relation to bile acid disposition. Within BIOLOGXsym, the impact of GGF2 on the regulation of hepatic enzyme and transporter expression was represented by generating a hepatic interstitial GGF2-concentration-driven signal acting upon individual bile acid and bilirubin disposition-related mechanisms (e.g., upregulation or downregulation of synthesis, uptake, efflux, or amidation).

### 4.14. Setup of GGF2-Mediated Hepatotoxicity Simulations in BIOLOGXsym

Proof-of-concept predictions of biologics-induced liver injury responses for GGF2 in BIOLOGXsym were made by integrating the mechanistic LAMPS with published data, and via hepatic GGF2 exposure predicted by the simulation of clinical protocols within the GastroPlus PBPK model. Simulated hepatic biomarker (i.e., ALT, total bilirubin) responses in BIOLOGXsym were compared with clinical data. Sensitivity analyses were performed in the baseline (i.e., “average”) human by varying the magnitude of GGF2 effects on enzyme/transporter expression to explore the impact of uncertainty in parameter values. Additional simulations were conducted in a normal healthy volunteer SimCohorts (n = 16 simulated individuals representing varying levels of patient susceptibility), composed of the baseline human, 12 subjects with sensitivity to bile acid transport inhibition as well as oxidative stress and mitochondrial dysfunction, and 3 subjects with low sensitivity in these areas.

## 5. Conclusions

In this study, BIOLOGXsym, a mechanistic model representing the physiological processes involved in biologics-induced liver injury was developed in conjunction with assay outputs from a liver microphysiology system. Phenotypic and mechanistic data and metabolomics analysis from the LAMPS model demonstrated that tocilizumab and GGF2 increased hepatic injury and stress markers, that tocilizumab exposure is associated with increased oxidative stress, and that GGF2 decreased bile acid secretion. Proof-of-concept simulations leveraging in vivo exposure predicted by PBPK modeling and mechanistic toxicity data from the LAMPS reproduced the clinically observed liver signals of tocilizumab and GGF2 [5,7], demonstrating that in vitro data from a microphysiology system can be successfully integrated into a QST model to recapitulate biologics-induced liver injury liabilities and provide mechanistic explanation for observed clinical liver safety signals. Additional analysis on a wider variety of biologic drugs is necessary to further refine the BIOLOGXsym simulation as a predictive tool to prospectively evaluate biologics-induced liver injury liabilities.

## Figures and Tables

**Figure 1 ijms-24-09692-f001:**
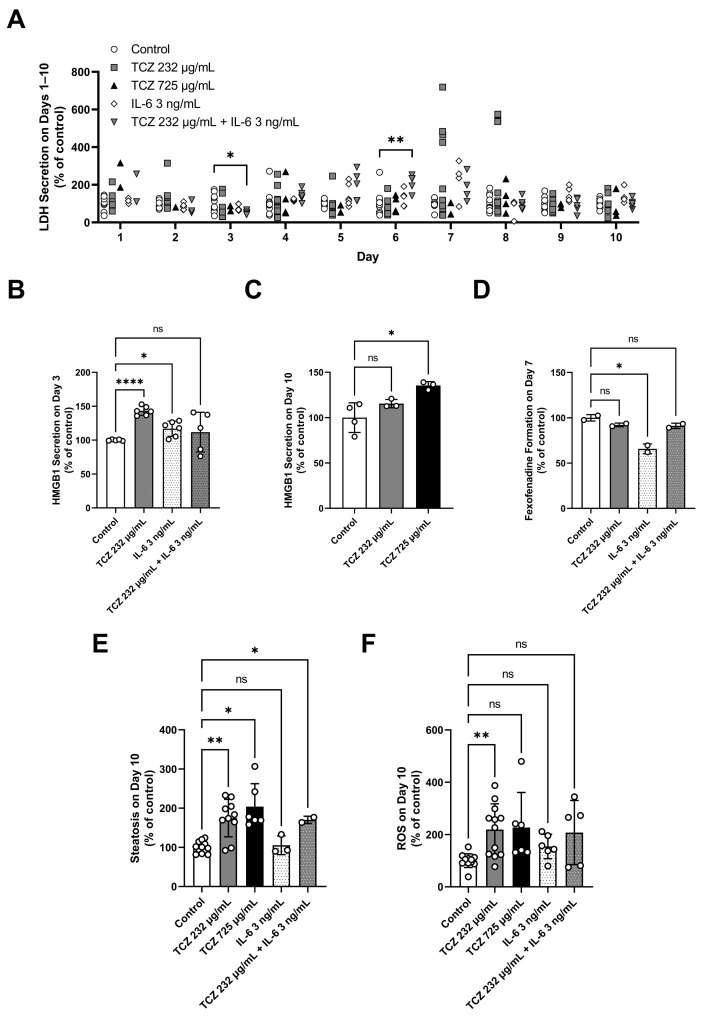
Significant toxicity findings in LAMPS chips treated with tocilizumab (232 µg/mL and 725 µg/mL), IL-6 (3 ng/mL), or a combination of tocilizumab (232 µg/mL) and IL-6 (3 ng/mL). (**A**) LDH on days 1–10, (**B**) HMGB1 on day 3, (**C**) HMGB1 on day 10, (**D**) CYP3A4-mediated formation of fexofenadine from terfenadine on day 7, (**E**) steatosis, as measured by LipidTox, on day 10, and (**F**) ROS, as measured by dihydroethidium, on day 10. n = 15, 14, 6, 6, and 5 chips for control, 232 µg/mL tocilizumab, 725 µg/mL tocilizumab, IL-6, and tocilizumab + IL-6 groups, respectively. Most endpoints were measured for a subset of chips tested. Values presented as mean ± SD. Statistical significance is indicated by asterisks; * < 0.05, ** < 0.01, **** < 0.0001. ns, not significant. CYP, cytochrome P450; HMGB1, high mobility group box 1; IL-6, interleukin-6; LAMPS, Liver Acinus MicroPhysiology System; LDH, lactate dehydrogenase; ROS, reactive oxygen species; TCZ, tocilizumab.

**Figure 2 ijms-24-09692-f002:**
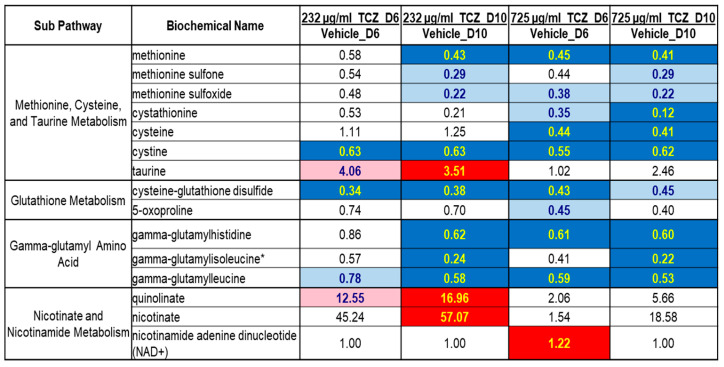
Statistical heat map, pathway diagram, and boxplots of select metabolites associated with oxidative stress within the spent media from the LAMPS models. Within the heatmap, trending (0.05 < *p* < 0.10) and significant (*p* ≤ 0.05) elevations are indicated by pink and red, respectively, while trending and significant reductions are represented by light blue and dark blue, respectively. * The compound’s chemical identity was confirmed by its chromatographic and spectral characteristics (RT and molecular fragmentation pattern) and mass, but not based on an authentic chemical standard for the metabolite. Box plots were used to demonstrate the distribution profile of a metabolite where the spread of the data with the middle 50% of the data represented by the box and the whiskers reporting the range of the data. The solid bar across the box represents the median value of that metabolite while the + represents the mean. Data are scaled such that the median value measured across all samples was set to 1.0.

**Figure 3 ijms-24-09692-f003:**
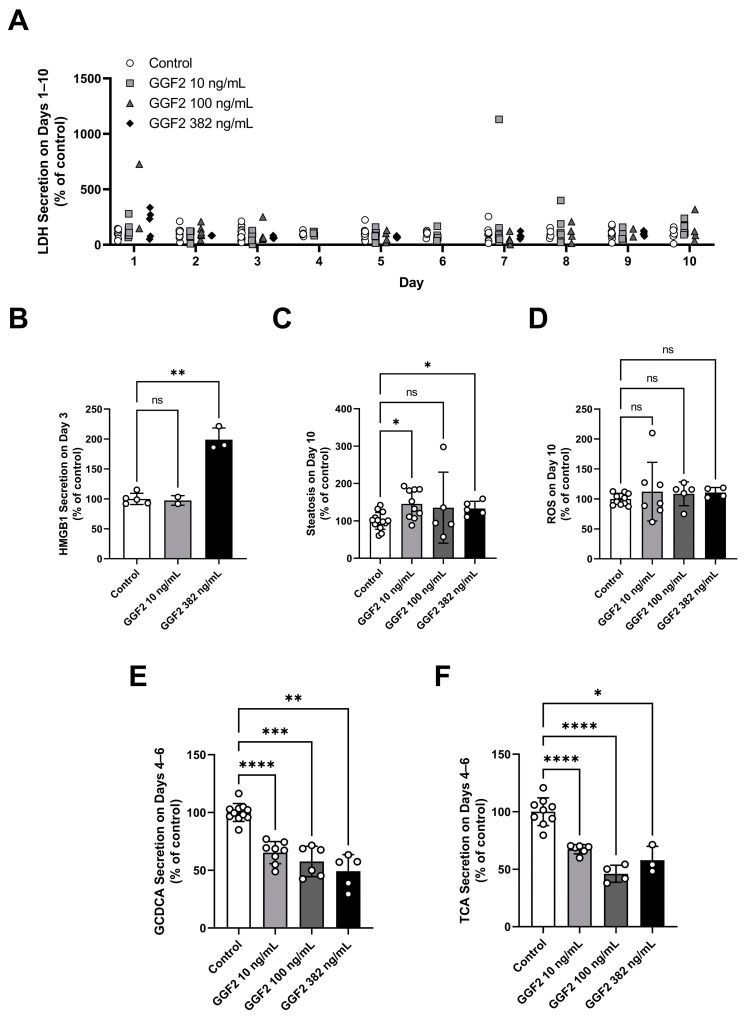
Significant toxicity findings in LAMPS chips treated with GGF2 at 10, 100, and 382 ng/mL. (**A**) LDH on days 1–10, (**B**) HMGB1 on day 3, (**C**) steatosis on day 10, (**D**) ROS on day 10, (**E**) glycochenodeoxycholic acid on days 4–6, and (**F**) taurocholic acid release on days 4–6. n = 15, 11, 6, and 5 chips for control, 10 ng/mL, 100 ng/mL, and 382 ng/mL groups, respectively. Some endpoints were measured for a subset of chips tested. Values presented as mean ± SD. Statistical significance is indicated by asterisks; * < 0.05, ** < 0.01, *** < 0.001, **** < 0.0001. ns, not significant. GCDCA, glycochenodeoxycholic acid; HMGB1, high mobility group box 1; LAMPS, Liver Acinus MicroPhysiology System; LDH, lactate dehydrogenase; ROS, reactive oxygen species; TCA, taurocholic acid.

**Figure 4 ijms-24-09692-f004:**
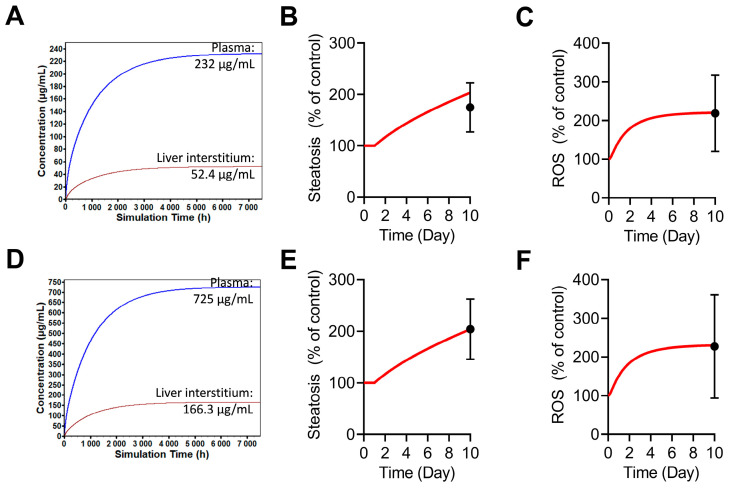
In-vitro-like simulations of tocilizumab mimicking LAMPS experiments. GastroPlus^®^ PBPK modeling of tocilizumab was used to simulate steady-state plasma concentrations that match the dosing concentrations of tocilizumab in the LAMPS experiments ((**A**): 232 µg/mL; (**D**): 725 µg/mL) to subsequently predict the corresponding hepatic interstitial concentrations. Hepatic interstitial concentrations were used to drive the steatosis (**B**,**E**) and ROS accumulation (**C**,**F**) parameterization in BIOLOGXsym. Red profiles represent simulated profiles in BIOLOGXsym, whereas solid black circles with error bars represent the LAMPS data (mean ± SD). LAMPS, Liver Acinus MicroPhysiology System; PBPK, physiologically based pharmacokinetic; ROS, reactive oxygen species; TCZ, tocilizumab.

**Figure 5 ijms-24-09692-f005:**
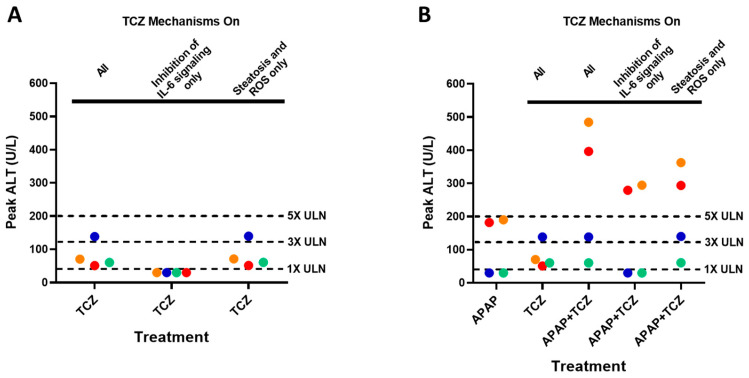
Simulated peak ALT responses in the SimCohorts (n = 4) administered (**A**) tocilizumab alone (8 mg/kg IV given every 4 weeks for 12 weeks), (**B**) acetaminophen alone (1 g four times a day (4 g/day), 12 weeks), tocilizumab alone (8 mg/kg IV given every 4 weeks for 12 weeks), or acetaminophen + tocilizumab. The impact of the tocilizumab-mediated inhibition of explicitly modeled IL-6 signaling and tocilizumab-mediated steatosis and ROS elevations was explored for tocilizumab alone and acetaminophen + tocilizumab simulations. Each symbol represents each simulated individual; symbol color corresponds to same individual. Dotted horizontal lines indicate ULN multiples (1×, 3× and 5×) of peak ALT, with ALT ULN defined as 40 U/L. ALT, alanine aminotransferase; APAP, acetaminophen; IL-6, interleukin-6; TCZ, tocilizumab; ULN, upper limit of normal.

**Figure 6 ijms-24-09692-f006:**
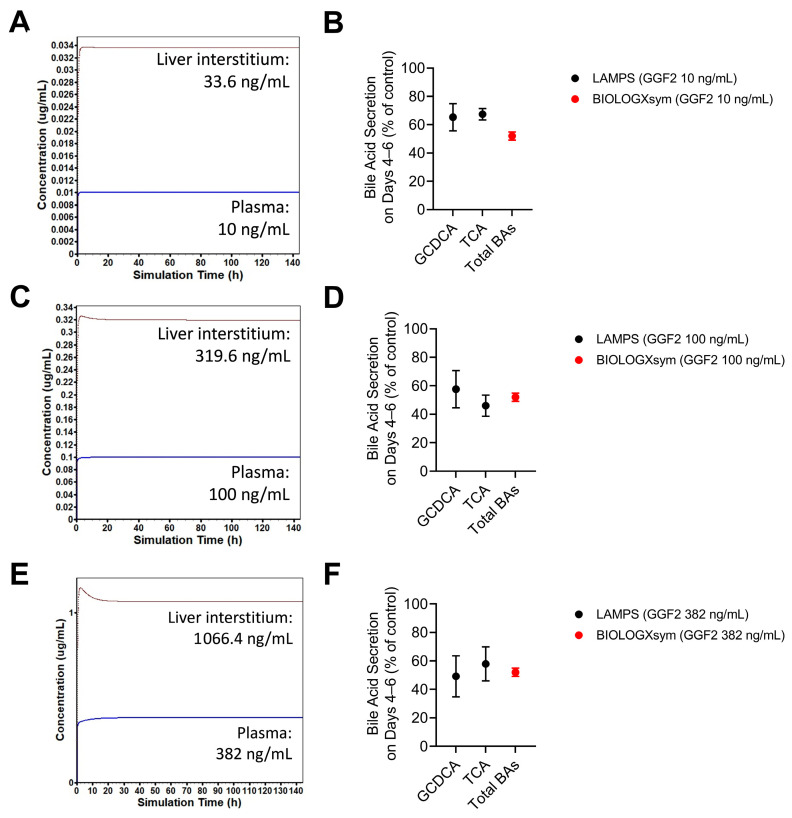
In-vitro-like simulations of GGF2-mimicking LAMPS experiments. GastroPlus PBPK modeling of GGF2 was used to simulate steady-state plasma concentrations that match the dosing concentrations of GGF2 in the LAMPS experiments ((**A**): 10 ng/mL; (**C**): 100 ng/mL; (**E**): 382 ng/mL) to subsequently predict the corresponding hepatic interstitial concentrations. Hepatic interstitial concentrations were used to drive the mechanisms underlying altered bile acid secretion (**B**,**D**,**F**) parameterization in BIOLOGXsym. Red solid circles with error bars represent simulated values (mean ± SD) in BIOLOGXsym on days 4, 5 and 6, whereas solid black circles with error bars represent the LAMPS data (mean ± SD) on these days. GCDCA, glycochenodeoxycholic acid; TCA, taurocholic acid; BAs, bile acids.

**Figure 7 ijms-24-09692-f007:**
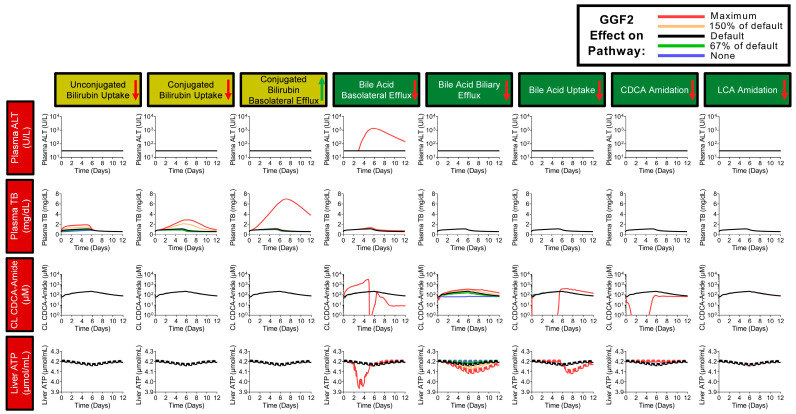
Sensitivity analysis of GGF2 effects on bilirubin and bile acid disposition-related mechanisms in the baseline human. Red and green arrows in the headers indicate down- or upregulation of the respective pathway based on transcriptional data [4]. The default toxicity parameter values across all panels are based on the magnitude of GGF2 impact on each individual mechanism using the transcriptional data and were subsequently decreased or increased as indicated in the color-coded legend. ALT, alanine aminotransferase; ATP, adenosine triphosphate; CDCA, chenodeoxycholic acid; CL, centrilobular; TB, total bilirubin.

**Figure 8 ijms-24-09692-f008:**
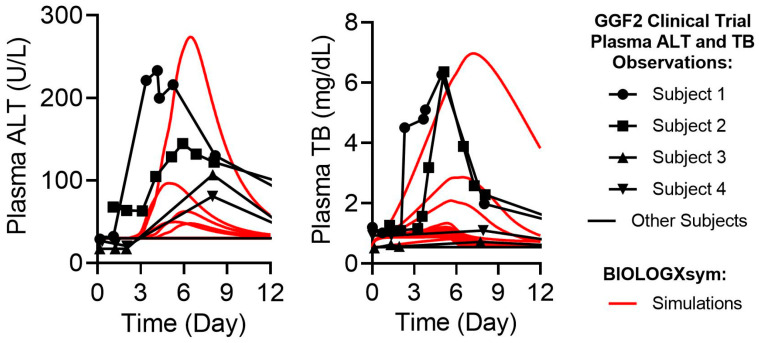
Comparison of simulated (red lines) and observed (black symbols and lines) GGF2-mediated hepatic responses. Solid red lines represent plasma ALT and total bilirubin simulation results in the SimCohorts (n = 16). Plasma total bilirubin simulation results in the baseline human with varying magnitudes of GGF2 effects on bilirubin transporters are also presented to account for to account for uncertainty around transcriptional data. ALT, alanine aminotransferase; TB, total bilirubin.

**Table 1 ijms-24-09692-t001:** Allometric scaling of LAMPS model.

Cell Type	Seeding Density	Final Cell Count (% Total)	Scaling Reference
Hepatocytes	2.75 × 10^6^/mL	74,000 (60%)	[69,70]
LX2 stellate	1.4 × 10^5^/mL	6000 (5%)	[69,70]
THP1 Kupffer Cells	8 × 10^5^/mL	18,000 (15%)	[69,70]
LSEC	1.5 × 10^6^/mL	24,000 (20%)	[69,70]

## Data Availability

The data and summary results are accessible through the internet at biosystics-ap.com. Data that have been released for public access are viewable to anyone visiting the site. The data for the five studies presented in this report will be made accessible to the public upon publication of the article. All use of the MPS-Db and content is free for non-profit research applications provided any published works reference the MPS Database website. All registered users have free access to the public content and tools; however, use of the data or tools for-profit applications requires a license. For more information, send an email to MPSHELP@pitt.edu with “For-profit Use of the MPS-Db” in the subject line.

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
