# Peer review of "The Combination of a Human Biomimetic Liver Microphysiology System with BIOLOGXsym, a Quantitative Systems Toxicology (QST) Modeling Platform for Macromolecules, Provides Mechanistic Understanding of Tocilizumab- and GGF2-Induced Liver Injury"

_ijms, 2023, doi:10.3390/ijms24119692_

Round 1
Reviewer 1 Report
The study is interesting. The authors should use the template of the Journal and follow its instructions.
Author Response
We thank the reviewer for the helpful comment. The manuscript has been updated using the template of the International Journal of Molecular Sciences. The formatting changes have not been tracked so that reviewers can easily find content changes.
Reviewer 2 Report
Hepatoxicity is a common complication when patients are treated with newly developed biologics. The development of cimaglermin alfa (GGF2) as drug intended for the treatment of heart failure was terminated in phase 1 because of signs of hepatotoxicity. Tocilizumab, which was developed for the treatment of rheumatoid arthritis, also induced hepatotoxicity as indicated by transient aminotransferase elevations and routine liver tests were recommended before, during and after the treatment period. Unfortunately, for the time being, there is no reliable in vitro model that is capable of predicting the degree of hepatotoxicity of novel biologics. In the present study authors explored the impact of two biological drugs (tocilizumab, cimaglermin) on liver injury using the BIOLOGXsym modeling platform in combination with mechanistic toxicity data obtained in an in vitro liver acinus microphysiology system. They found that the in vitro data can be integrated into the BIOLOGXsym modeling platform and that the updated program successfully predicts the observed clinical liver injury readout parameter. Nevertheless, the authors concluded that additional analyses on a wider variety of biologic drugs is required to further refine the BIOLOGXsym simulation as a predictive tool to prospectively evaluate the potential hepatotoxic properties of novel biological drugs.
The ms is on an important topic and is of direct clinical relevance. The data are well presented and I think the ms will frequently read not only be pharmacologists but also by clinicians who are using biologicals for the treatment of diseases. However, the ms is extremely difficult to read because of the excessive use of non-standard abbreviations. This might not be a problem for specialized pharmacologist but for average readers this excessive used of standard abbreviations severely hampers legibility of the paper. Although the authors spell out all abbreviations when they first appear in the text but it is almost impossible for an average reader to keep in mind the meaning of an abbreviation found on page 10 of the ms that was spelt out on page 1. I have no idea how to solve this problem but the authors should find a way to reduce the non-standard abbreviations to an absolute minimum. Sometimes it is better to spell them out throughout the ms. For those non-standard abbreviations that cannot be avoided an abbreviation list should be included in the ms. Non-expert readers can refer to this list anytime the find an abbreviation in the tex. The Abstract should be free of abbreviations. They should all be spelt out even if they repeatedly occur.
Author Response
We thank the reviewer for the helpful feedback. To address the reviewer’s comment and improve the readability by broader audiences, we have 1) removed most of non-standard abbreviations and spelled them out throughout the main text, 2) removed all abbreviations from the abstract except for the drug name (GGF2), and 3) added a list of abbreviations at the end of the manuscript (Lines 803-813 in the revised manuscript).
Reviewer 3 Report
This work reports a model system LAMPS along with a computational model powered by BIOLOGXsym that work together to allow for examination of BILI and prediction of the effects of different treatments on BILI. The authors have presented thorough and promising data. I however have a few suggestions and questions that I believe could help with the manuscript's clarity.
1. I think it would be helpful to provide more details on how BIOLOGXsym performs the simulations. For example, is it stochastic or continous simulations? What solver and settings it uses to integrate the system?
2. In Table S1 and S2, a few parameters were optimized. How was this done? Was it by fitting the entire or a part of model to the data? If so, what is the algorithm used to fit the data? How well was the fit? I think it would be great to provide more technical details on this.
3. Are observed data variations incorporated into the model? If so, how was it done?
